# IFITM3 Clusters on Virus Containing Endosomes and Lysosomes Early in the Influenza A Infection of Human Airway Epithelial Cells

**DOI:** 10.3390/v11060548

**Published:** 2019-06-12

**Authors:** Susann Kummer, Ori Avinoam, Hans-Georg Kräusslich

**Affiliations:** 1Department of Infectious Diseases, Virology, Heidelberg University Hospital, Heidelberg 69120, Germany; 2Department of Biomolecular Sciences, Weizmann Institute of Science, Rehovot 7610001, Israel

**Keywords:** influenza A virus, IFITM3, super-resolution microscopy, viral-host interaction, endosomes/lysosomes

## Abstract

Interferon-induced transmembrane proteins (IFITMs) have been shown to strongly affect influenza A virus (IAV) infectivity in tissue culture. Moreover, polymorphisms in IFITM3 have been associated with the severity of the disease in humans. IFITM3 appears to act early in the infection, but its mechanism of action and potential interactions with incoming IAV structures are not yet defined. Here, we visualized endogenous IFITM3 interactions with IAV in the human lung epithelial cell line A549 and in primary human airway epithelial cells employing stimulated emission depletion super-resolution microscopy. By applying an iterative approach for the cluster definition and computational cluster analysis, we found that IFITM3 reorganizes into clusters as IAV infection progresses. IFITM3 cluster formation started at 2-3 h post infection and increased over time to finally coat IAV-containing endosomal vesicles. This IAV-induced phenotype was due to the endosomal recruitment of IFITM3 rather than to an overall increase in the IFITM3 abundance. While the IAV-induced IFITM3 clustering and localization to endosomal vesicles was comparable in primary human airway epithelial cells and the human lung epithelial cell line A549, the endogenous IFITM3 signal was higher in primary cells. Moreover, we observed IFITM3 signals adjacent to IAV-containing recycling endosomes.

## 1. Introduction

Influenza A virus (IAV) is the major cause for a contagious illness of the upper and lower respiratory tract during the seasonal influenza epidemics with peaks in fall and winter for each hemisphere [1,2,3]. Besides the acute risk through circulating human specific strains, the zoonotic reservoir (e.g., birds, swine) poses a constant threat of new influenza pandemics [4,5,6,7].

IAV hijacks cellular import mechanisms to enter the host cell, thereby making use of multiple entry routes. The binding of IAV by the sialylated receptor [8,9] is followed by clathrin-mediated endocytosis [10,11] or micropinocytosis [12]. It is reported that IAV entry is cell-type dependent [13] and that subtypes with a filamentous particle shape preferentially enter host cells via macropinocytosis [14]. The trimeric surface glycoprotein hemagglutinin (HA) is a key factor for several steps during viral entry via endocytosis [15,16]. Irrespective of the entry mechanism, IAV exploits the non-linear endosomal pathway with its multitude of branches and needs to pass different stages of the endocytic machinery, which is assembled and constantly renewed around the internalized virus particles [17,18,19]. Endosomal trafficking to the perinuclear region is crucial for pH-dependent membrane fusion [20,21,22], followed by the release of the viral genome into the cytosol and nuclear import [23]. The IAV genome is composed of eight single RNA strands in a negative sense orientation [24], and each segment is complexed with the nucleoprotein NP [15,24], forming ribonucleoparticles (RNPs; [25,26]). Viral genome replication, splicing and transcription then occur in the nucleus. IAV replication is strongly inhibited by type 1 interferons already at the early stage, with Mx1 being identified as a crucial interferon-induced host protein blocking IAV replication [27].

Genome-wide siRNA screens have identified many host factors modulating influenza virus infection [28,29,30,31,32]. Brass et al., 2009 [32] and Shapira et al., 2009 [33] reported the strong inhibition of IAV infection by interferon-induced transmembrane proteins (IFITMs). The IFITM variants 1–3 can be induced by interferon I and II. Their localization is cell type- or tissue-dependent and changes with the expression level with a preference for cytoplasmic vesicles in most monolayer cell lines [34,35,36,37,38]. Besides IAV, IFITM proteins have also been reported to confer a basal resistance to members of the Flaviviridae (Dengue and West Nile Virus) [39], Bunyaviridae (Rift Valley Fever Virus) [40], Filoviridae (Ebola Virus) [41], Coronaviridae (SARS) and Retroviridae (HIV) [32], showing an unusually broad activity against a wide variety of enveloped and some non-enveloped viruses [32]. However, murine leukemia virus (MLV) is not restricted by IFITM3 despite being an enveloped virus [32]. The relevance of the IFITM3-mediated inhibition of IAV infection received strong support when it was shown that IFITM3 polymorphisms correlated with the severity of IAV disease in human infection [42]. The relevance of the IFITM3 rs12252-C polymorphism for severe IAV infection appears to be population-dependent [43,44,45,46,47,48], and a second IFITM3 single nucleotide polymorphism, rs34481144-A, was also reported to influence the severity of IAV infection in humans [49]. However, the mechanism of the IFITM-mediated inhibition of IAV infection remains under discussion.

IFITMs have been suggested to disrupt viral membrane fusion [50,51] by altering cellular membrane properties such as fluidity and curvature [38,52,53]. It has also been discussed that IFITMs alter the lipid/protein composition of acidic intracellular compartments such as endosomes and lysosome [37,54]. The lipid composition-based models of the IFITM3 function cannot easily explain the lack of antiviral effects on viruses like amphotropic MLV or arenaviruses [55,56], which also enter by endocytosis and endosomal fusion. In the case of IAV, it was recently reported that IFITM3 elevates the level of cholesterol on late endosomes and lysosomes, thereby restricting early IAV infection [57]. Other hypotheses suggest that IFITM3 directly interferes with the IAV fusion pore formation or redirects IAV containing endosomes to a non-productive pathway [41]. It was further demonstrated that the inhibition of the hemagglutinin-mediated membrane fusion required the amphipathic helix of IFITM3 [58]. Most findings about the antiviral action and localization of IFITM3 are based on studies using expression plasmids creating an over-expression and/or focusing at the late stages of infection (>12 h post-infection) [37,38,51]. Due to the enormous protein load in the overexpression situation, it is far more difficult to determine the subtle differences in the localization of individual molecules, and thus it is possible that these changes may be overlooked. In addition, proteins whose expression is enhanced artificially are more prone to be degraded via the lysosomal pathway. In this regard, it is difficult to judge whether this is a natural re-localization as part of the antiviral defense or just a degradation via lysosomes, particularly in the late stages of infection. We therefore investigated the role of endogenous IFITM3 in the viral uptake at an early stage.

We analyzed the localization of endogenous IFITM3 through the course of IAV infection using confocal and super-resolution (STED) microscopy. We observed a strong clustering of IFITM3 early after IAV infection, which was distinct from the interferon alpha induced IFITM3 up-regulation. Moreover, a two-color STED nanoscopy revealed a close proximity of IFITM3 with the IAV NP protein in Rab11 positive compartments within a time range of 1–6 h after the IAV addition.

## 2. Materials and Methods

### 2.1. Cells, Reagents, Viruses and Plasmids

A549 and MDCK cells were maintained in Dulbecco`s modified Eagle Medium (DMEM) (Invitrogen, Karlsruhe, Germany), supplemented with 10% foetal calf serum (FCS) (Invitrogen) and penicillin (200 U/mL)/streptomycin sulphate (100 µg/mL) (Capricorn scientific GmbH, Ebsdorfergrund, Germany). Human small airway epithelial cells (HSAEpCs) were obtained from PromoCell (C-12642) and grown in a primary cell basal medium (PromoCell) without antibiotics. Four days after thawing, the HSAEpCs were seeded for immunofluorescence. The IFITM3 knock-down in A549 cells was induced by CRISPR-Cas9 gene editing using a single guide RNA targeting the polyadenylation side of IFITM [59,60]. A549 IFITM3 knock-down cells were selected by puromycin resistance (10 µg/mL). The IFITM3 expression was checked by real-time quantitative PCR, as described by Zhao et al., 2013 [61] using the 7500 Fast Real Time PCR System (Applied Biosystems, Lifetechnologies, sigma-aldrich, Steinheim, Germany) and by immunofluorescence, as noted below (Appendix A).

In general, the cells subjected to immunofluorescence were seeded in 24 well plates on 10 mm glass cover slips. For immunoblotting, cells were seeded in 6 well plates.

Human interferon alpha (hIFNα) (sigma-aldrich, Steinheim, Germany) was used at 1000 U/mL for 24 h prior to infection. Amantadine (sigma-aldrich) and Bafilomycin A1 (Merck, Darmstadt, Germany) were applied to the cells at 5 µM and 10 nM concentrations 30 min prior to infection. The cells were treated with mouse anti-hIFNα (catalogue no. ab11408, abcam, Berlin, Germany).

Influenza A virus A/Hong Kong/1/1968 (HK/1/68), A/Puerto Rico/8/1934 (PR/8/34) and A/Regensburg/D6/2009 (R/D6/09) were recovered in a HEK 293T-MDCK-coculture using the eight plasmid reverse genetics systems, as previously described [62]. In live-cell experiments, Cell Mask™ Orange plasma membrane stain (catalogue no. C10045, Life Technologies, sigma-aldrich, Steinheim, Germany) was directly added to the cell supernatant prior to the virus purification, following the manufacturer`s guidelines. The virus stocks were titrated, making use of a standard plaque assay with minor changes, as previously described [63]. Briefly, serial 10-fold dilutions of each virus stock were prepared in DMEM supplemented with 0.3% bovine serum albumin (BSA, sigma-aldrich, Steinheim, Germany), 20 mM HEPES (sigma-aldrich, Steinheim, Germany), penicillin (200 U/mL)/streptomycin sulphate (100 µg/mL) (Capricorn scientific GmbH, Ebsdorfergrund, Germany) and TPCK-trypsin (2 µg/mL) (sigma-aldrich, Steinheim, Germany). MDCK cell monolayers were incubated with 200 µL of each dilution for 1 h at 37 °C in a 6 well plate format. Then, the cells were overlaid with 2 mL of 2.4% Avicel (FMC) in aqua dest. and 2 × DMEM (Merck, Darmstadt, Germany) in a 1:1 ratio. After 5 days, the overlay was removed, the cells were fixed with 1 mL/well of 70% ethanol for 30 min at room temperature and stained with 1 mL/well of 0.3% crystal violet (sigma-aldrich, Steinheim, Germany) in 20% methanol for 10 min at room temperature.

### 2.2. Immunofluorescence Staining

The cells were fixed using 4% paraformaldehyde (PFA) in 1 × phosphate buffered saline (PBS) for 10 min at 37 °C and subsequently treated with 0.05% Triton-X100 in PBS for 5 min. Blocking was performed using 5% BSA in PBS for 10 min. The following primary antibodies were used for immunofluorescence: mouse anti-NP (catalogue no. MAB8257, Merck, Darmstadt, Germany), rabbit anti-IFITM3 (catalogue no. 11714-1-AP, Proteintech, Manchester, United Kingdom), mouse anti-EEA1 (catalogue no. 610457, BD Transduction LaboratoriesTM, San Jose, CA, U.S.A.), mouse anti-Rab7 (catalogue no. ab50533, abcam, Berlin, Germany), mouse anti-Rab11 (catalogue no. sc-6565, Santa Cruz, Dallas, Texas, U.S.A.), and mouse anti-lamp1 (catalogue no. ab25630, abcam, Berlin, Germany). The cells were incubated with the indicated primary antibodies in a blocking solution for 1 h at room temperature, and washed three times in PBS. The fixation, cellular membrane permeabilisation, and blocking were repeated between the first and secondary antibody incubation, as described above. For STED microscopy, goat anti-mouse STAR RED (catalogue no. 2-0002-011-2, abberior, Göttingen, Germany) and goat anti-rabbit Atto594 (catalogue no. ABIN964988, antibodies-online GmbH) were used as the secondary antibodies. For the triple staining, including the non-diffracted 488 nm channel, donkey anti-goat Alexa 488 (catalogue no. ab150129, abcam, Berlin, Germany) was used. The secondary antibody staining was performed for 1 h at room temperature under exclusion of light. After repeated washing in PBS, cover slips were mounted on the objective slides using Mowiol.

### 2.3. Immunoblotting (Western Blotting)

Whole-cell lysates were prepared using a 6 × lysis buffer (10% sucrose, 0.1% bromophenol blue, 5 mM EDTA pH 8.0, 200 mM Tris pH 8.8, 10% SDS, and 2% beta- mercaptoethanol). Protein samples were resolved by 12.5% SDS-PAGE at 160 Volt for 1.5 h. Then, the proteins were transferred to Immobilon-FL membranes (Merck, Darmstadt, Germany). The membranes were blocked in Li-Cor blocking buffer for 30 min at room temperature and probed with first antibodies (rabbit anti-IFITM3, catalogue no. 11714-1-AP, Proteintech, Manchester, United Kingdom, and rabbit anti-actin, catalogue no. A2066, sigma-aldrich, Steinheim, Germany, mouse anti-tubulin no. T5168, sigma-aldrich, Steinheim, Germany, or mouse anti-GAPDH, catalogue no. sc365062, Santa Cruz, Dallas, Texas, U.S.A.) in a blocking solution diluted 1:5000 at 8 °C overnight. After three repeated washings in 1 × TBST for 15 min, the membranes were incubated with a secondary donkey anti-mouse 800 antibody (catalogue no. 926-32212, Li-Cor) or secondary donkey anti-mouse 700 antibody (catalogue no. 925-68072, Li-Cor, Lincoln, Nebraska U.S.A.), for 1 h at room temperature in the dark. The protein bands were visualized using the Li-Cor Odyssey scanner (Li-Cor, Lincoln, Nebraska U.S.A.).

### 2.4. Microscopy and Data Analysis

Fluorescence imaging was performed using a two-colour-STED microscopy instrumentation (Abberior Instruments GmbH, Göttingen, Germany). For confocal and STED microscopy, a 100× Olympus UPlanSApo (NA 1.4) oil immersion objective was used. For excitation (λ= 594 nm and λ = 640 nm), a nominal laser power of 20% was applied, and for STED (λ = 775 nm, max. power = 1.2 W), a nominal laser power of 70 % was applied. The pixel size was set to 60 nm (confocal) and 15 nm (non-diffracted), respectively. Minor adjustments of contrast and brightness of the acquired images as well as the Richardson-Lucy deconvolution with a regularisation parameter of 0.001 (stopped after 30 iterations) were carried out with the imspector software (16.1.7098-win64-AIFpgaV3, Abberior Instruments GmbH, Göttingen, Germany).

The 2D object counter and coloc2 plugin in Fiji software (https://fiji.sc/, 64-bit version)were used for the cluster and Mander’s Coefficient Correlation (MCC) analyses, respectively. MCC was performed using the Costes regression threshold. The defined objects were assigned as a cluster being larger than 10^5^ nm^2^ based on STED images with approximately 50 nm resolution. All graphs show the mean values of three independent experiments plotted with the standard error calculated as a two tailed unpaired *t*-test. A statistical analysis was performed using GraphPad Prism software (version 5).

## 3. Results

### 3.1. IFITM3 Is Re-Distributed upon IAV Infection

To visualize the subcellular localization of endogenous IFITM3, we performed immunofluorescence microscopy of the uninfected and IAV infected A549 cells. The basal IFITM3 levels of the uninfected cells showed weak cytosolic signals upon immunofluorescence staining. Following infection with IAV A/Hong Kong/1/1968 (HK/1/68) for 24 h, most cells exhibited strong and clustered IFITM3 signals in the extranuclear space (Figure 1A). The IFITM3 signal increase and clustering correlated with the overall dim NP signals in the extranuclear space and no nuclear NP signals in the respective cells. In contrast, the productively IAV infected A549 cells (exhibiting a strong nuclear NP signal) in the same field of view did not exhibit an increased IFITM3 signal intensity or clustering (Figure 1A, right panels). Given that we used a relatively high MOI of one and that dim NP signals were generally detected in these cells, these observations strongly argue that IAV infection was abortive in the cells exhibiting an early accumulation of IFITM3.

IFITM3 is induced by type 1 interferon [34]. Therefore, we analyzed its distribution upon the treatment of A549 cells with human interferon alpha (hIFNα) for 24 h and compared this to A549 cells which had been pretreated with hIFNα and subsequently infected with IAV (Figure 1B). The cells treated for 24 h with hIFNα exhibited an increased IFITM3 signal intensity in the extranuclear space, both in the uninfected A549 cells and in the IAV-infected cells with a dim or absent NP signal. Importantly, the productively infected cells exhibiting strong nuclear NP signals again exhibited a weak IFITM3 signal, even after the pretreatment with hIFNα. Comparing the non-treated and hIFNα-pretreated IAV-infected A549 cells lacking a strong nuclear NP signal, we observed an increased IFITM3 signal in both cases, partially colocalizing with vesicular structures. Vesicular localization appeared more obvious in the absence of a hIFNα pretreatment with a less diffuse cytosolic signal. Blocking IFNα with a neutralizing antibody during the IAV infection of A549 cells resulted in increased infection rates, and a concomitant minimal increase of the IFITM3 signal (Figure 1C).

To determine whether an IAV-induced viral membrane fusion and genome uncoating are required for the observed IFITM3 signal increase upon IAV infection, we performed experiments in the presence of Bafilomycin A1, specifically inhibiting endosomal acidification, or in the presence of Amantadine, specifically blocking the tetrameric M2 channel of IAV, thereby preventing genome uncoating. In both cases, no increase in the IFITM3 signal intensity and no IFITM3 clustering were observed (Figure 1D,E). These results indicate that IAV-induced membrane fusion and genome uncoating are required for an IFITM3 signal increase in A549 cells.

The mechanism by which IFITM3 impedes IAV infection was divergent in previous studies, and inhibition was mostly described upon IFITM3 over-expression [37,38,51]. It was demonstrated that an IFITM3 knockdown in A549 cells increases infection rates [37]. To determine whether endogenous IFITM3 is relevant for IAV infection in A549 cells under the experimental conditions of our study, we performed IAV infection in A549 IFITM3 knock-down cells (Appendix A). The IFITM3 expression was reduced by a factor of 10 (on the mRNA level) in these cells (Appendix A). The IAV infection rate was significantly increased in the knock-down situation compared to the A549 wildtype cells (Figure 2A). 

The increase in the IFITM3 signal intensity in A549 cells upon IAV infection (Figure 1A) and interferon treatment (Figure 1B) could be caused either by higher expression levels or by IFITM3 clustering, yielding a higher density of the signal. To distinguish between these possibilities, we determined the IFITM3 abundance by an immunoblot analysis of the A549 cells at different time points after IAV infection or interferon treatment, respectively. No significant increase in the IFITM3 expression was observed during the first 6 h of IAV infection (Figure 2B), indicating that the signal changes observed by fluorescence microscopy (examples are shown in Figure 3) during this period were caused by the re-localization of constitutively expressed IFITM3 rather than by an induced IFITM3 expression. A strong increase in the IFITM3 expression levels was observed at late time points (24 h p.i.; Appendix A), in accordance with previous studies [32]. An increased IFITM3 expression was also observed following the interferon treatment, but starting already at 6 h after the interferon addition (Appendix A), as previously reported [32]. Accordingly, the immunofluorescence phenotype after the interferon treatment (Figure 1B) is likely to be caused by an increased expression rather than by clustering.

### 3.2. STED Analysis Reveals IFITM3 Clustering in IAV Infected A549 and Primary Human Respiratory Epithelial Cells

To unravel the distribution pattern of IFITM3 in A549 cells at early stages of infection, we made use of super-resolution microscopy. The blurring effect of diffraction limited imaging can obfuscate the potential clustering of IFITM3. Stimulated emission depletion (STED) super-resolution microscopy was therefore applied to resolve IFITM3 clusters in early IAV-infected A549 cells. Cells infected with IAV HK/1/68 were fixed at 1–6 h post infection (h p.i.), stained with anti-IFITM3 and subjected to confocal and STED microscopy (resolution < 50 nm) of the same region (Figure 3A). An increased IFITM3 signal intensity was again observed in the early phase of the IAV infection, and individual IFITM3 clusters could be resolved by STED microscopy. We computationally analysed the IFITM3 subcellular localization based on raw STED images using a 2D cluster analysis (objects counter). The clusters were defined as extranuclear IFITM3 signal accumulations with a size of > 10^5^ nm^2^. The threshold was chosen by an iterative approach searching for the proportion of clusters, with the critical size being altered upon the infection progression.

The analysis of uninfected and HK/1/68-infected A549 cells at different time points revealed a very low proportion of large clusters in uninfected cells or up to 2 h p.i. At 3 h p.i. and later, we observed a significant increase in cluster abundance (Figure 3B, left graph). A similar induction of IFITM3 clusters was also observed upon infection with the IAV strain A/Regensburg/D6/2009 (R/D6/2009, H1N1), but this occurred with slower kinetics (Figure 3B, right graph). The same phenotype was also observed in IAV A/Puerto Rico/8/1934 (PR/8/34)-infected A549 cells (Susann Kummer, Heidelberg, Germany, A549 cells stained for IFITM3 and NP, 2019).

Although A549 cells are an established model cell line for IAV research, adaptation to cell-culture conditions may have occurred; hence, we verified our results in primary human respiratory epithelial cells. To determine whether IAV infection also causes IFITM3 clustering in primary cells, we infected human small airway epithelial cells (HSAEpCs) with IAV PR/8/34 for different periods of time and performed an indirect immunofluorescence analysis for IFITM3 and NP (from incoming IAV particles) using confocal and STED microscopy (Figure 4, uninfected example see Appendix A). The IAV infected HSAEpCs revealed a co-localization of IFITM3 and IAV NP signals at apparently vesicular structures as early as 1 h p.i. (Figure 4A, white arrowheads). At later time points (3 h p.i.), this was more obvious, with a clear IFITM3 clustering on the NP-containing vesicular structures (Figure 4B). IFITM3 often exhibited a ring-like appearance, suggesting the coating of endosomal vesicles, and this phenotype became more obvious at later time points (6 h p.i; Figure 4C). Some IFITM3-positive vesicles exhibited strong NP signals (e.g., Figure 4B), suggesting IFITM3-coated vesicles carrying multiple IAV particles.

A strong IFITM3 clustering with a ring-like appearance indicating vesicle coating was observed in both IAV-infected A549 cells (Figure 5A) and HSAEpCs at 10 h p.i. (Figure 5C; white arrowheads indicating ring-like structures). In the case of the A549 cells, the IFITM3 signals in uninfected cells were weak and mostly diffuse. Upon IAV infection, > 90% of cells lacking a strong nuclear NP signal displayed IFITM3 clustering (e.g., the right cell in Figure 5A), while this was only observed in < 10% of cells with a strong nuclear NP (Figure 3B). Given the high infection rate in this cell line and the fact that cytoplasmic NP was also observed in cells lacking the bright nuclear NP of replicating IAV (Figure 1A), we suggest that early IFITM3 clustering at vesicular structures correlates with an abortive infection in this cell type, and that only cells that do not induce the vesicular clustering of IFITM3 become infected. In general, the IFITM3 signals were stronger in HSAEpCs compared to A549 cells (Figure 5A,C), and IFITM3 clusters apparently coating cytoplasmic vesicles were also present in productively IAV-infected cells exhibiting a strong nuclear NP signal in this case (Figure 5D). No IFITM3 clustering at vesicular structures was observed in HSAEpCs in the absence of IAV infection, while the overall IFITM3 signal intensity was much higher in these primary cells compared to A549 cells without infection (Appendix A).

### 3.3. IFITM3 Is Recruited to Endosomes at Early Stages of IAV Infection

The ring-like appearance of IFITM3 clusters suggested the vesicular recruitment of this protein to IAV-carrying endosomal structures. We therefore analysed whether IFITM3 clusters co-localize with early endosomal (EEA1), late endosomal (Rab7), or lysosomal (Lamp1) marker proteins at early stages of IAV infection in A549 cells (Figure 6, Appendix A). We observed typical staining for early and late endosomes and for lysosomes with a weak IFITM3 signal in uninfected cells. The IFITM3 signal intensity increased at 1 h and 6 h p.i., as seen in the previous experiments, with some apparent co-localization with the endosomal marker proteins. To determine the degree of co-localization, we calculated the Mander’s Correlation Coefficient (MCC) [64]. Only IFITM3 clusters defined as objects with a size of > 10^5^ nm^2^ were included in this analysis. We observed a highly significant increase of IFITM3 cluster co-localization with early endosomes at 1 h p.i., which was subsequently lost in the course of infection (Figure 6A). In contrast, a significant increase of co-localization with the late endosomal marker (Figure 6B) and some increased co-localization with the lysosomal marker (Figure 6C) was seen at 6 h, but not at 1 h p.i. These results suggest that IFITM3 clustering on IAV carrying endosomes already occurs early after endocytosis, prior to the pH-induced fusion and release from late endosomes, and probably persists as the endosomes mature.

### 3.4. IFITM3 Is Present on IAV Containing Recycling Endosomes

Endocytosed material is generally sorted to different destinations: reusable ligands and receptors are returned to the cell surface via recycling endosomes, while the material destined for degradation traffic to lysosomes [65]. Previous studies reported a substantial co-localization between virus-containing compartments and Rme-1 that is thought to be associated with sorting and recycling endosomes [66]. Furthermore, recent reports have suggested that Rab11-associated recycling endosomes are required for vRNP trafficking, assembly, and virion budding at the cellular surface in the late phase of IAV infection [67,68,69,70].

To determine whether IAV and IFITM3 clustering can be detected on recycling endosomes, we compared uninfected and IAV-infected A549 cells at different time points. Some co-localization of NP with IFITM3 and Rab11 signals was observed on apparently vesicular structures in the extranuclear space of IAV infected A549 cells at 1–6 h p.i. (Figure 7A–C). However, the co-localization of IFITM3 and Rab11 was already present in the uninfected A549 cells, with no significant change of co-localization during the course of infection, as shown by the MCC analysis (Appendix A). The uptake of IAV particles into recycling endosomes is expected to lead to an abortive infection, as recycling endosomes do not undergo acidification during trafficking. It is likely that rare events of an enhanced IFITM3 and Rab11 co-localization are underestimated in the bulk MCC analysis algorithm, as indicated by the MCC value fluctuation (Appendix A). 

## 4. Discussion

The IFITM3-mediated inhibition of IAV infection has been suggested to be caused by changes in the membrane tension resulting in a block of fusion pore formation [71], and may be due to constitutively expressed protein or interferon induction [38,41,57,72]. Applying confocal and super-resolution imaging of endogenous IFITM3, we show that pre-existing IFITM3 clusters at early and late endosomal structures carry IAV at early time points of infection, prior to interferon induction. As shown previously [32] and in our study (Appendix A), interferon alpha led to increased protein levels of IFITM3 at later time points. On the other hand, no increase in the IFITM3 protein levels was observed up to 6 h p.i. in IAV infected A549 cells.

A semi-automated image analysis of IAV A/HK/1/68 (pdmH3N2) and IAV A/R/D6/2009 (pdmH1N1) infected A549 cells revealed IFITM3 clustering early after infection in both cases, with few differences in kinetics, which may reflect strain specific replication kinetics [73]. The CD225 domain of IFITM3, which is composed of the intermembrane domain 1 and the intracellular loop, contains two phenylalanine residues mediating the physical association between IFITMs; these residues are strongly connected with the antiviral function [39]. Combining this finding with our observation of the increasing levels of larger IFITM3 clusters, we suggest a direct relation between cluster formation and IFITM3`s antiviral activity on endosomal vesicles. The IFITM3 clustering on apparently vesicular structures started early after infection (3–5 h p.i.), with an initial co-localization of IFITM3 with an early endosomal marker and a later co-localization with a late endosomal marker, reflecting the early to late endosomal pathway of IAV. It appears likely, therefore, that IFITM3 clusters, initially recruited to early endosomes and then coating endosomal vesicles through their trafficking pathway, may mediate the antiviral activity of IFITM3 and block the release of vRNPs. This hypothesis is further supported by the identification of a critical sorting signal that is essential for the IFITM3 localization to endosomes and its anti-viral activity [74,75]. Similar to IAV, arena- (including LASV, lymphocytic choriomeningitis virus, and MACV) and alphaviruses (including the chikungunya virus, Sindbis virus, and Venezuelan encephalitis virus) also fuse from late endosomes and lysosomes at a similar acidic pH, but are not restricted by IFITM3 [51]. It will be of interest, therefore, whether a similar IFITM3 clustering can be observed on endosomal vesicles carrying these viruses or whether they can escape from IFITM3 clustering.

Importantly, a similar phenotype was observed in human primary respiratory cells from healthy donor tissue (HSAEpCs). The basal levels of IFITM3 were much higher in these primary cells, compared to A549 cells in the absence of IAV infection. However, IFITM3 clustering was not observed in the naïve primary cell population, but rapidly increased again upon IAV infection and was even stronger than in A549 cells. IFITM3 clustering on cytoplasmic vesicles was strongly induced in both A549 cells and HSAEpCs exhibiting weak extranuclear signals for IAV NP, which most likely reflects an incoming input virus. In contrast, no such IFITM3 clusters were seen in productively infected A549 cells with a strong nuclear NP signal, and we speculate that infection of this cell type mainly occurs in cells that have not managed to block IAV entry by inducing IFITM3 clustering. Alternatively, IFITM3 clusters may be abrogated once productive infection has occurred and may thus no longer be visible in these cells. Live-cell microscopy using labelled IFITM3 will be required to distinguish between these possibilities. A different phenotype was observed in primary HSAEpCs, where IFITM3 clustering on vesicular structures was also observed in productively infected cells with a strong nuclear signal. Despite having a much higher basal level, IFITM3 clustering in HSAEpCs does not appear, therefore, to efficiently block IAV infection in these primary cells.

From our findings we assume two putative and presumably additive mechanisms for IFITM3 mediated antiviral action: (i) The first contact with IAV particles after an endocytic uptake in (early) endosomes (role of recycling endosomes is discussed below) triggers the continuous recruitment of pre-existing IFITM3 proteins to IAV containing compartments as a fast antiviral defense in the initial phase of infection. (ii) IFITM3 protein levels are increased by interferon signaling, further propagating its antiviral effect and possibly synergizing with additional IFN-induced antiviral factors [76]. Continuing studies aim to identify the modalities of the direct or indirect interaction between IAV and IFITM3. We have to admit that deficient particles might have the potential to fuse with the plasma membrane but fail to replicate. An abortive and ineffective infection might trigger the IFITM3 activation and recruitment to endosomes in the same way as can be seen for an efficient IAV infection.

Various members of the endosomal network fulfil crucial functions in the viral entry, trafficking, and genome release. Rab11 is mainly assigned to recycling endosomes [67]. We made the observation that Rab11-positive vesicles loaded with IFITM3 often carried IAV NP early during infection. Previous reports using live-cell imaging suggested that IFITM3 mediates the directed re-localization of viral particles to lysosomes [77]. Our results argue that the internalization of IAV via recycling endosomes may trap IAV in a non-productive pathway, as the membrane fusion is precluded in recycling endosomes having only a mildly acidic pH of 6.5 [78]. It is conceivable that both mechanisms exist synergistically. In general, viruses being restricted by IFITM3 fuse at a lower pH in late endosomes or lysosomes [51]. IFITM3 on recycling endosomes is thus unlikely to restrict the endosomal entry of IAV (or of other viruses described to be restricted by IFITMs), but may be an important defense against other pathogens entering via recycling endosomes [78].

## Figures and Tables

**Figure 1 viruses-11-00548-f001:**
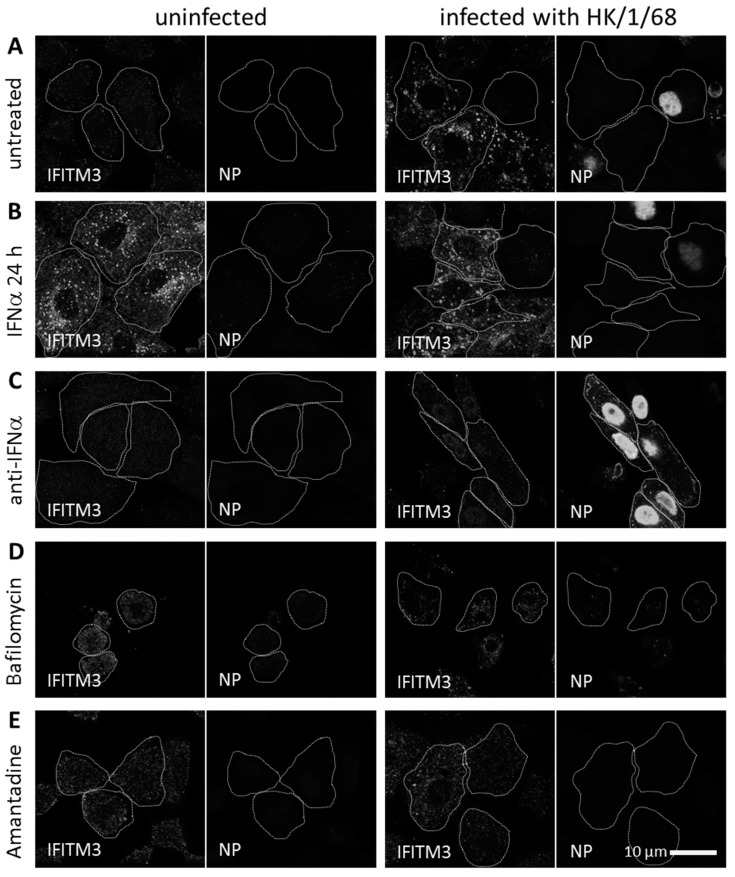
IFITM3 cluster formation in influenza A virus (IAV) infected A549 cells. The indirect immunofluorescence of (**A**) untreated, (**B**) IFNα treated for 24 h, (**C**) anti-human IFNα treated for 24 h, (**D**) Bafilomycin A1 treated, and (**E**) Amantadine treated A549 cells, using anti-IFITM3 and IAV anti-nucleoprotein antibodies. The non-infected control (left panels) and IAV A/Hong Kong/1/1968 (MOI = 1) infected A549 cells (right panels) were fixed with PFA at 24 h p.i. The images shown represent the major subcellular distribution of the respective protein. The periphery of the immunostained cells was outlined with white dotted lines on the basis of the cell morphology. The image acquisition was performed using a 60 nm (= confocal mode) pixel size.

**Figure 2 viruses-11-00548-f002:**
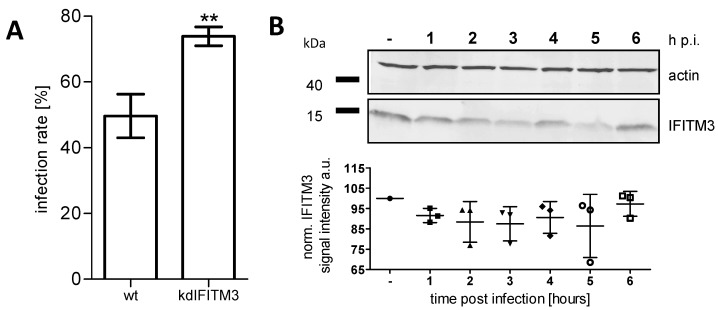
The effect of the IFITM3 knock-down on the influenza A virus (IAV) infection rate and the effect of IAV infection on IFITM3 expression levels. The A549 wildtype and IFITM3 knock-down cells subjected to immunofluorescence were infected with IAV A/Hong Kong/1/1968 (MOI = 1) and fixed at 8 h p.i. The cells were stained using an anti-IFITM3 antibody in combination with anti-nucleoprotein. The image acquisition was performed using a 60 nm (= confocal mode) pixel size. (**A**) The infection rate of A549 wildtype and IFITM3 knock-down cells was calculated based on immunostaining against IAV NP at 8 h p.i. Cells harboring nuclear NP signals were scored as IAV infected. The mean values of three independent experiments are plotted as the mean (SD). (**B**, top) The immunoblot analysis of the IFITM3 levels in whole-cell lysates from non-infected (-) and A/Hong Kong/1/1968 infected A549 wildtype cells (times post-infection, as indicated). Actin was used as the loading control. The panels are derived from a single cropped blot, with lanes 1–7 belonging to the same membrane and from the same experiment. ** *p* = 0.0044 (**B**, bottom) The signal intensities of IFITM3 signals normalized for the actin control of three independent experiments. The mean values are plotted as the mean (SD). The IFITM3 levels are non-significantly different from each other.

**Figure 3 viruses-11-00548-f003:**
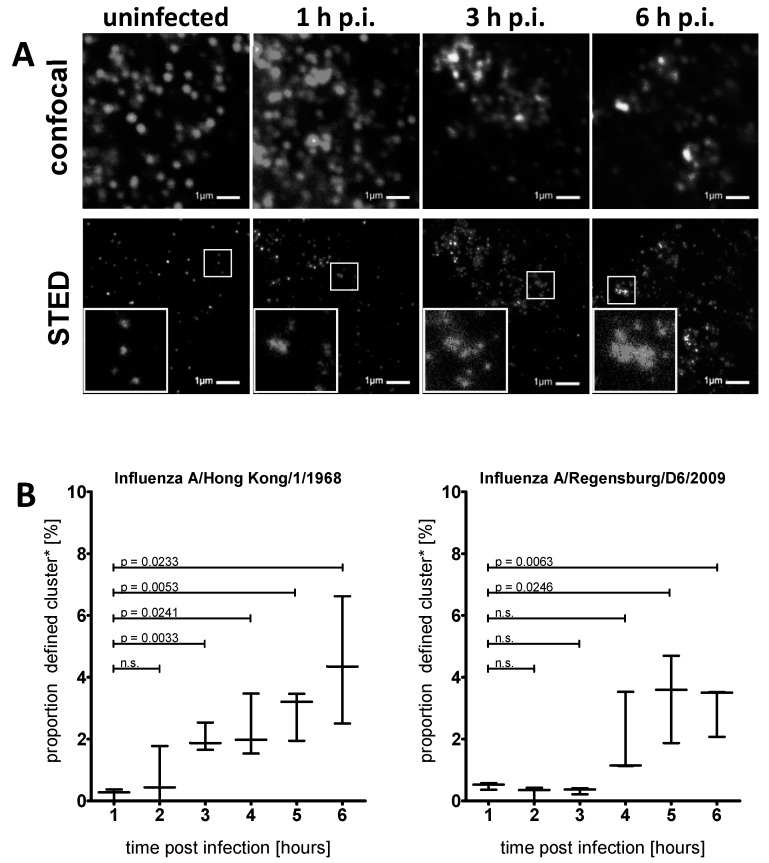
IFITM3 clusters in influenza A virus (IAV) infected A549 cells. (**A**) Indirect immunofluorescence using confocal (upper panel) and STED (lower panel) analyses of IAV A/Hong Kong/1/1968 (MOI = 1) infected A549 cells using an anti-IFITM3 antibody. The STED images (raw data) show representative subcellular regions located in the cytosolic part, distant from the nucleus and absent from the plasma membrane, as used for the cluster analysis. The white rectangles show zoom-ins of structures, defined as clusters. Image acquisition with 60 nm (= confocal mode, top) and 15 nm (= STED mode, bottom) pixel size. (**B**) The analysis of the cluster size of A/Hong Kong/1/1968 infected (left panel) and A/Regensburg/D6/2009 infected (right panel) A549 cells using the objects counter plugin from Fiji. The proportion of clusters in relation to all detected objects is plotted. The data are represented as a Whiskers plot showing minimum and maximum values and the median of three independent experiments (*n* = 3). Clusters are defined as objects larger than 10^5^ nm². n.s. = non significant.

**Figure 4 viruses-11-00548-f004:**
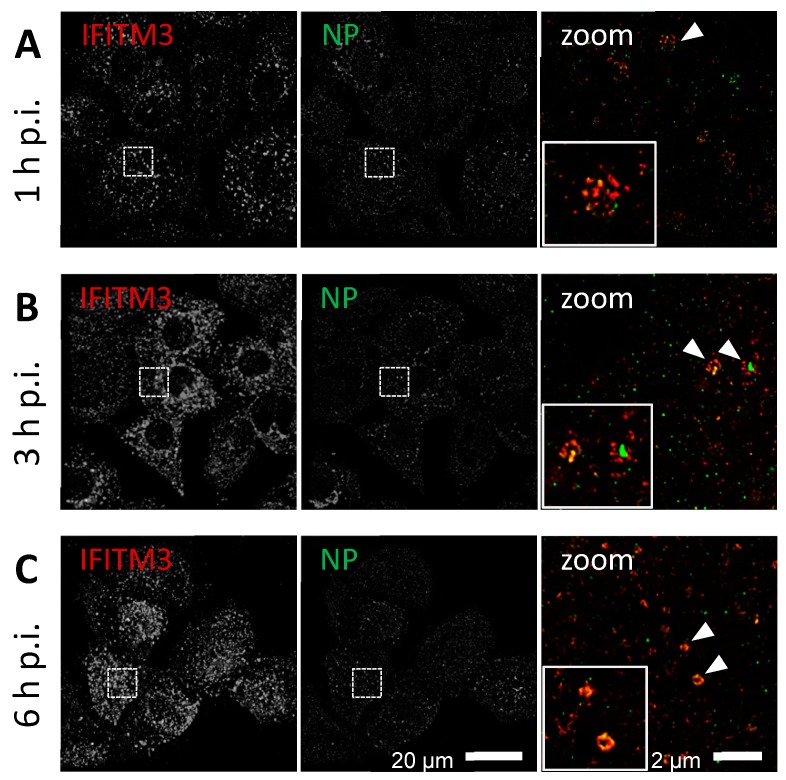
IFITM3 co-localization with influenza A virus (IAV) nucleoprotein (NP) in primary human cells. The indirect immunofluorescence analysis of human small airway epithelial cells (HSAEpCs) infected with IAV A/Puerto Rico/8/1934 (MOI = 10) using anti-IFITM3 (red) in combination with anti-NP (green). (**A**) 1 h p.i., (**B**) 3 h p.i., and (**C**) 6 h p.i. The images show extra-nuclear regions of HSAEpCs. The image acquisition was performed using a 60 nm (= confocal mode, overview) and 20 nm (= STED, merged zoom) pixel size. The dashed square indicates the zoom-in area. White arrow heads indicating ring-like structures with IFITM3 clustering.

**Figure 5 viruses-11-00548-f005:**
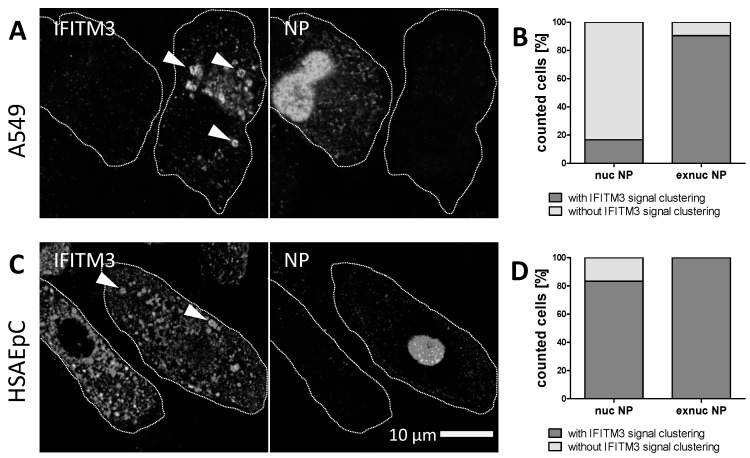
IFITM3 enriched regions in IAV infected cells. The indirect immunofluorescence of influenza A virus (IAV) A/Puerto Rico/8/1934 (MOI = 1) infected (**A**) A549 cells and (**C**) HSAEpCs. The cells subjected for fluorescence microscopy were fixed at 10 h p.i. and stained using anti-IFITM3 and anti-NP antibodies. The periphery of the immunostained cells was outlined with white dot lines on the basis of the cell morphology. White arrow heads point at vesicular structures. The image acquisition was performed using a 60 nm (= confocal mode) pixel size. (**B**,**D**) The analysis of the IFITM3 clustering with respect to the presence of nuclear (nuc) or extranuclear (exnuc) NP signals, respectively.

**Figure 6 viruses-11-00548-f006:**
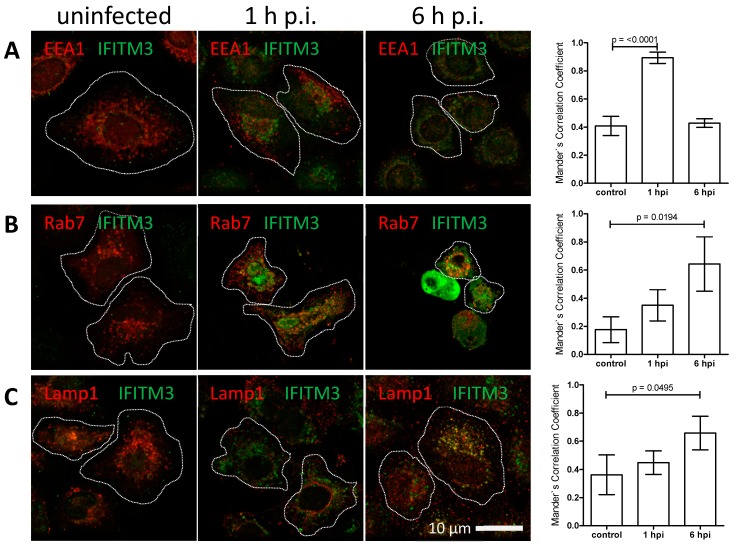
The co-localization of IFITM3 proteins with early and late endosomes and lysosomes. A549 cells were infected with influenza A virus (IAV) A/Hong Kong/1/1968 (MOI = 1) and fixed with ice cold methanol at the indicated time points. The indirect immunofluorescence analysis using anti-IFITM3 (green) in combination with (**A**) anti-EEA1 (red), (**B**) anti-Rab7 (red) and (**C**) anti-Lamp1 (red). The images show the merged acquisition channels. The periphery of the immunostained cells was outlined with white dotted lines on the basis of the cell morphology. The image acquisition was performed using a 60 nm (= confocal mode) pixel size. (right panel) The signal intensities for IFITM3 were correlated with (**A**,**B**) endosomal and (**C**) lysosomal markers. Mander’s Correlation Coefficients from three independent experiments for each condition were calculated using the coloc2 plugin implemented in Fiji software. The mean values are plotted with the mean (SD).

**Figure 7 viruses-11-00548-f007:**
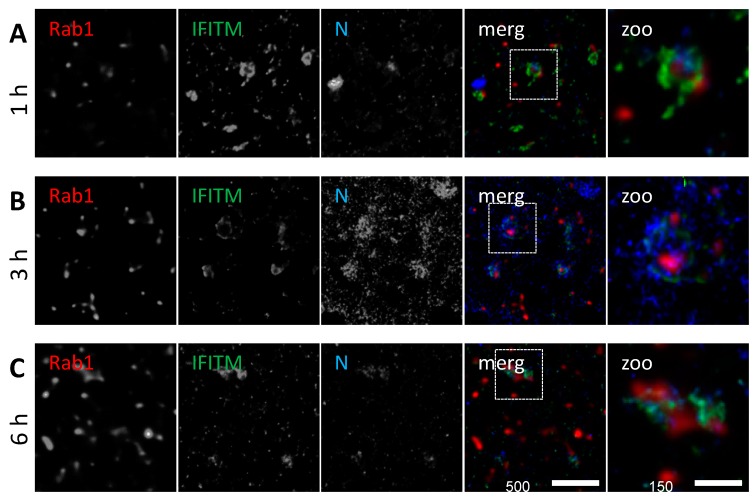
The co-localization of IFITM3 proteins with Rab11 and IAV NP. A549 cells were transiently transfected with pEGFP-C3_Rab11 (green) and infected with influenza A virus (IAV) A/Hong Kong/1/1968 (MOI = 1) 48 h after transfection. The cells were fixed with ice cold methanol at (**A**) 1 h p.i., (**B**) 3 h p.i., and (**C**) 6 h p.i., and subjected to an indirect immunofluorescence analysis using anti-IFITM3 (red) in combination with anti-NP (blue). The images show regions located in the cytosolic part, distant from the nucleus and absent from the plasma membrane. The image acquisition was performed using a 60 nm (= confocal mode, EGFP-Rab11) and 15 nm (= STED, IFITM3 and NP signal) pixel size. The dashed square indicates the zoom-in area.

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
