# Peer review of "IFITM3 Clusters on Virus Containing Endosomes and Lysosomes Early in the Influenza A Infection of Human Airway Epithelial Cells"

_viruses, 2019, doi:10.3390/v11060548_

Round 1
Reviewer 1 Report
In this study the authors examine the localization of IFITM3 during a viral infection in airway epithelial cells. They convincingly shown that IFITM3 clusters in cells where NP is primarily in the cytoplasm in an interferon dependent manner. Using super-resolution microscopy the authors show colocalization of IFITM3 with influenza viral nucleoprotein and early endosomal proteins. Overall it is an interesting study, using sophisticated microscopy techniques to examine the intracellular distribution of IFITM3 during viral infection. There are few clarifications that would strengthen the overall manuscript.
1. The largest concern is regarding the colocalization of IFITM3 and Rab11A. It seems (as the authors state) that these represent abortive events (Figure 7). Therefore the inclusion of the movie examining the colocalization of Rab11A GFP, IFITM3-SNAP, and labeled virion seems completely unnecessary. In addition, the movie does not provide any details on cellular features and a viewer is left wondering where the foci are coming from and going. This reviewer suggests removing the movie or alternatively including labels on the movie of cellular features and including a scale bar.
2. Additional minor comments on the figures include:
a. Supplemental figures should be included in the main text.
b. The NP stain in HSAEpCs is only shown to be cytoplasmic, inclusion of an uninfected control would demonstrate specificity of the staining. At later timepoints is the NP stain nuclear as expected, is this simply a delay in replication?
c. Labels or zoomed out images from those presented in Figure 3 and 7 are needed to orient the reader to where within a cell the colocalization is observed.
d. Figure 2B - Western blot examining levels of IFITM3 should include 24 hpi to correlate with the data presented in Figure 1.
Author Response
1. The largest concern is regarding the colocalization of IFITM3 and Rab11A. It seems (as the authors state) that these represent abortive events (Figure 7). Therefore the inclusion of the movie examining the colocalization of Rab11A GFP, IFITM3-SNAP, and labeled virion seems completely unnecessary. In addition, the movie does not provide any details on cellular features and a viewer is left wondering where the foci are coming from and going. This reviewer suggests removing the movie or alternatively including labels on the movie of cellular features and including a scale bar.
Following the suggestion of this reviewer, we have removed the movie in the revised version.
2. Additional minor comments on the figures include:
a. Supplemental figures should be included in the main text.
The supplemental figures only include experiments that reproduce previously published work or are not absolutely necessary to support the major and most novel findings of the paper. We therefore suggest to keep these data in the supplement. If this is not acceptable we will make the changes in final revision.
b. The NP stain in HSAEpCs is only shown to be cytoplasmic, inclusion of an uninfected control would demonstrate specificity of the staining. At later timepoints is the NP stain nuclear as expected, is this simply a delay in replication?
We show indirect immunofluorescence analysis with anti-IFITM3 and anti-NP antibodies of HSAEpCs in Figure S6. No unspecific signal is detected in the cytosol of HSAEpCs, indicating that the observed cytoplasmic NP stain is specific.
Nuclear NP stain is detected at 10 h p.i., and this may indeed indicate delayed replication compared to A549 cells as suggested by the reviewer, but no direct side-by-side comparison of replication kinetics in different cells was performed.
c. Labels or zoomed out images from those presented in Figure 3 and 7 are needed to orient the reader to where within a cell the colocalization is observed.
The IFITM3 images were acquired in the cytosol distant from the nucleus and absent from the plasma membrane. We included this information in the respective figure legend of Figure 3 and 7.
Figure 3: “STED images (raw data) show representative subcellular regions located in the cytosolic part distant from the nucleus and absent from the plasma membrane as used for cluster analysis.”
Figure 7: “Images show regions located in the cytosolic part distant from the nucleus and absent from the plasma membrane.”
d. Figure 2B - Western blot examining levels of IFITM3 should include 24 hpi to correlate with the data presented in Figure 1.
We now include the Western Blot analysis suggested by the reviewer as Figure S2B and thank the reviewer for suggesting this important control.
Reviewer 2 Report
This study provides imaging of the antiviral protein IFITM3 at endogenous levels and at super-resolution in the context of influenza virus infection. They show that IFITM3 clusters within 1 h of influenza virus infection, prior to increases in the protein’s levels as induced by the infection. These IFITM3 clusters were found to correlate with restriction of virus infection and to co-localize with incoming virus. Further, they observed that the IFITM3 clusters that form early in infection are at early endosomes and progress to late endosomes and lysosomes as infection proceeds, consistent with IFITM3 directing incoming virus to degradative compartments. While there are important new insights provided by this work, the manuscript can be improved in terms of data presentation and referencing the existing literature on the IFITM proteins.
Figure 1 should be labeled as to what treatments are used in panels A-E. As it is now, the figure on its own cannot possibly be interpreted without the legend.
Figure 2A: The siRNA control should be shown. qPCR is acceptable, but an IFITM3 Western would be better, particularly since the antibody is readily available.
Figure 2B: These data need to be improved. The IFITM3 blot shows wide variability with no pattern from one hour to the next, and yet the conclusion is that there is no change in IFITM3 levels during this time period.
Figure 3A: Why is there so much IFITM3 present in the uninfected cells? This does not appear to increase post infection as was shown previously.
Figure 4: Uninfected control should be shown in the main text.
Figure 4: Label the figure so that it can be interpreted independent of the legend.
Figure 5A/C: Label the figure so that it can be interpreted independent of the legend.
Figure 7: I am not sure that this figure is a useful addition to the paper as it suffers from exactly the problems that the authors claim to overcome in their study, i.e., the use of overexpressed IFITM3 and an overexpressed Rab11 marker protein, both of which may show unnatural localization due to the overexpression.
Lines 227/228: This sentence is misleading as the references cited are not contradictory, nor are they unclear as to IFITM3 possessing antiviral activity. Furthermore, the authors need to examine the literature more carefully, as previous studies have indeed shown that IFITM3 knockdown in A549 cells increases influenza virus infection, e.g., Lin, Cell Reports, 2013.
Since this paper is primarily studying IFITM localization and clustering at endosomes/lysosomes it should reference studies in which endocytic localization motifs of the IFITMs were identified and characterized (Jia, Cell Microbiol, 2014; Chesarino, JBC, 2014; Li, JBC, 2015)
Line 65 states that subsequent studies have failed to confirm an association between SNP rs12252 and severe influenza. This may be true, but far more studies have confirmed this link than have refuted it. Further, the studies which failed to find an association were generally performed in populations in which the SNP is almost non-existent. Additionally, a second SNP in the IFITM3 promoter has also been linked to severe flu. For a review of these studies, see Zani, Current Clin Microbiol Reports, 2018, though the primary articles should be cited.
Lines 67-73 The inhibition of virus membrane fusion by IFITM3 has been attributed to the presence of a palmitoylated amphipathic helix within IFITM3 (Chesarino, EMBO Reports, 2017). This helix and neighboring palmitoylation sites are among the most highly conserved residues among IFITMs from all species. Additionally, the role of cholesterol in IFITM3’s mechanism of action has been largely disproven by the field, and reference 52 seems to have been misused in line 73.
Author Response
Figure 1 should be labeled as to what treatments are used in panels A-E. As it is now, the figure on its own cannot possibly be interpreted without the legend.
We have made the suggested changes.
Figure 2A: The siRNA control should be shown. qPCR is acceptable, but an IFITM3 Western would be better, particularly since the antibody is readily available.
We now include an immunoblot showing IFITM3 levels in whole-cell lysates from A549 wildtype and IFITM3 knock-down cells as Figure S1D.
Figure 2B: These data need to be improved. The IFITM3 blot shows wide variability with no pattern from one hour to the next, and yet the conclusion is that there is no change in IFITM3 levels during this time period.
To address the reviewer’s concern, we performed statistical analysis of three independent experiments shown in Figure 2B, lower panel. Statistical analysis of mean values of the determined protein amount for IFITM3 shows no statistically significant differences (e.g. p-value of 6 h pi compared to control is 0.5224).
We included this information in the figure legend of Figure 2.
Figure 3A: Why is there so much IFITM3 present in the uninfected cells? This does not appear to increase post infection as was shown previously.
Figure 2B shows that the amount of IFITM3 is not significantly altered up to 6 h p.i. IFITM3 protein increased significantly after longer infection times such as 24 h p.i. (Fig. S2B), as was previously described by others.
Figure 4: Uninfected control should be shown in the main text.
We have not made this change as we show that the anti-NP antibody is highly specific and shows negligible signal in the uninfected controls in Figure 1A. A representative example of non-infected HSAEpC is shown in the supplement. If this is not acceptable we will make the changes in final revision.
Figure 4: Label the figure so that it can be interpreted independent of the legend.
We have made the requested changes.
Figure 5A/C: Label the figure so that it can be interpreted independent of the legend.
We have made the requested changes to figure 5A/C and Figure 7.
Figure 7: I am not sure that this figure is a useful addition to the paper as it suffers from exactly the problems that the authors claim to overcome in their study, i.e., the use of overexpressed IFITM3 and an overexpressed Rab11 marker protein, both of which may show unnatural localization due to the overexpression.
We used a fluorescently tagged Rab11 because the anti-Rab11 antibodies were not sufficient to label recycling endosomes. However, IFITM3 is detected at native levels and this compromise allowed us to evaluate its localization. Overexpressed IFITM3 was only used for the live cell microscopy shown in the movie, which has now been removed (see reply to reviewer 1).
Lines 227/228: This sentence is misleading as the references cited are not contradictory, nor are they unclear as to IFITM3 possessing antiviral activity. Furthermore, the authors need to examine the literature more carefully, as previous studies have indeed shown that IFITM3 knockdown in A549 cells increases influenza virus infection, e.g., Lin, Cell Reports, 2013.
Based on the reviewer´s suggestion, we have now included again the reference showing the infection increase of IAV in A549 IFITM3 knockdown cells and have changed the wording to make the statement clearer.
“The mechanism by whichIFITM3 impedes IAV infection was divergent in previous studies and inhibition was mostly described upon IFITM3 over-expression [37, 38, 51]. It was demonstrated that IFITM3 knockdown in A549 cells increases infection rate [37].”
Since this paper is primarily studying IFITM localization and clustering at endosomes/lysosomes it should reference studies in which endocytic localization motifs of the IFITMs were identified and characterized (Jia, Cell Microbiol, 2014; Chesarino, JBC, 2015; Li, JBC, 2015)
All the references have been added and we now include this aspect of the IFITM3 mechanism in the discussion of the paper. We thank the reviewer for the comment.
Line 65 states that subsequent studies have failed to confirm an association between SNP rs12252 and severe influenza. This may be true, but far more studies have confirmed this link than have refuted it. Further, the studies which failed to find an association were generally performed in populations in which the SNP is almost non-existent. Additionally, a second SNP in the IFITM3 promoter has also been linked to severe flu. For a review of these studies, see Zani, Current Clin Microbiol Reports, 2018, though the primary articles should be cited.
Lines 61 ff. were changed pointing out the relevance of SNP rs12252-C and SNP rs34481144-A. References have been added.
Lines 67-73 The inhibition of virus membrane fusion by IFITM3 has been attributed to the presence of a palmitoylated amphipathic helix within IFITM3 (Chesarino, EMBO Reports, 2017). This helix and neighboring palmitoylation sites are among the most highly conserved residues among IFITMs from all species. Additionally, the role of cholesterol in IFITM3’s mechanism of action has been largely disproven by the field, and reference 52 seems to have been misused in line 73.
We thank the reviewer for pointing out the inaccuracy in the references. The statement "...IFITM3 elevates the level of cholesterol on late endosomes and lysosomes thereby restricting early IAV infection" is based on the publication of Kühnl et al. 2018 (mBio). The reference was replaced.
We have also complemented the introductory part with the aspect of fusion inhibition through the amphipathic helix of IFITM3.
Round 2
Reviewer 2 Report
The authors have largely addressed my previous comments.